# Stroking Rates of Open Water Swimmers during the 2019 FINA World Swimming Championships

**DOI:** 10.3390/ijerph18136850

**Published:** 2021-06-25

**Authors:** Luis Rodríguez, Santiago Veiga, Iker García, José M González-Ravé

**Affiliations:** 1Catalonian Swimming Federation, Diputació St., 237, 08007 Barcelona, Spain; dtnatacio@natacio.cat; 2Faculty of Sports Sciences, Universidad de Castilla-La Mancha, Carlos III Avenue, 45008 Toledo, Spain; 3Health and Human Performance Department, Universidad Politécnica de Madrid, Martin Fierro St., 28024 Madrid, Spain; 4Physiology Section, Department of Cell Biology, Physiology and Immunology, Faculty of Biology, Universitat de Barcelona, Av. Diagonal 643, 08028 Barcelona, Spain; ikergarciaalday@gmail.com; 5High Performance Center, Alcalde Barnils, Av. 3-5, Sant Cugat del Vallès, 08174 Barcelona, Spain; 6Sports Training Laboratory, Faculty of Sports Sciences, Universidad de Castilla-La Mancha, Carlos III Avenue, 45008 Toledo, Spain; josemaria.gonzalez@uclm.es

**Keywords:** competition, cadence, pacing, endurance, tactics

## Abstract

The aim of the present research was to examine the stroking rate (SR) values of successful and non-successful swimmers in the 10 km and 25 km races of the FINA 2019 World Swimming Championships. Data from 175 participants (95 men and 80 female) were classified according to their finishing positions. There were no meaningful differences in the overall SR values displayed by successful or non-successful participants during the 10 km and 25 km open water races of the FINA 2019 World Swimming Championships. However, there were changes in the SR throughout the races that depended on the swimmer’s performance group and gender. Successful swimmers in the 10 km event typically displayed even SR in the first 5 km but, unlike the remaining performance groups, increased their SR at some point in the second 5 km of the race. In the 25 km race, successful female swimmers presented an even SR profile for most of the race, whereas successful males presented a more variable profile. Nevertheless, no relationships between partial or average SR and finishing positions occurred, either in the 10 km or in the 25 km race. Changes in the SR values should be included in the race plan of open water swimmers according to tactical and pacing strategies.

## 1. Introduction

It is extensively acknowledged that optimal pacing (or energy output distribution) represents an indispensable ability for success in all endurance sport disciplines [1]. In open water swimming, since this has been an official event of the Olympic programme in 2008, there has been a growing interest in the scientific community about the race strategies of successful and non-successful participants [2]. It has already been described that pacing strategies and tactical behaviour of open water swimmers during 5 km, 10 km and 25 km races depend on the event distance [3,4] and on the swimmer’s finishing positions [3,5,6], with some specific features according the swimmer’s gender [6].

Successful open water swimmers in 10 km races (the event distance included in the Olympic programme) have been reported to employ overall swimming paces above the lactate threshold velocities [5]. They also generally present a conservative race strategy, consisting in swimming at a lower-than-average pace for the majority of the race distance, with delayed positioning within the main group. Medallists in these events are able to increase their swimming pace up to 5% in the last lap of the race distance [4], unlike other performance groups, who maintain or even decrease their swimming paces. The drafting effect could partly explain this conservative behaviour, as swimmers try to swim closely behind or at the side of another competitor to reduce their swimming energy cost by 20% [3,7]. Therefore, successful swimmers aim for delayed positions that can allow them to save energy for the last quarter of the race distance. This “low-profile” attitude of successful competitors within the main group (with gap times with respect to leaders of 15–20 s maximum) [4] could also allow them to avoid close contact with other competitors when disputing the leading positions. In the 25 km open water event, similar race strategies have been observed, with successful competitors swimming the second half considerably faster than the first half of the race [4]. However, in this event, the 5 h duration of the race emphasizes greater pacing variations due to athletes managing body temperature regulation [8] and also reacting to the group behaviour, rather than their maximal possible energy output [9]. This is what is has been referred to as herd behaviour in endurance disciplines [10].

During the last FINA World Swimming Championships held in Gwangju (South Korea) in July 2019, the Organizing Committee provided for the first time, in the public domain, the stroking rate (SR) values of the open water swimmers. The swimming SR can be defined as the number of strokes per minute or the time required for the swimmer to complete one stroke cycle [11]. It is a key parameter in the control of swimming intensity that can predict the performance [12], energy cost [13], technical ability [14,15] and fatigue level of swimmers [16]. In terms of gender comparisons, female swimmers present a similar SR to males in a variety of distances [17,18], although their shorter stroking length implies a greater dependence of women on SR in order to swim fast [19]. Also, young swimmers from 11 years of age are expected to present similar SR as senior swimmers, as the individual rhythm of swimming movement is established at that age [20]. Generally speaking, swimmers of a higher level of expertise are able to swim at a fixed speed with lower levels of SR (and, consequently, longer stroke lengths) than non-experts. It has been found that median SR (0.77 cycles per second = 46.2 cycles/min) optimizes the relationship between average velocity and VO_2_ compared to 0.68 (=40.8 cycles/min) or 0.87 (=52.2 cycles/min) cycles per second [21]. However, in endurance disciplines, when fatigue develops and the anaerobic threshold is overcome, swimmers tend to reduce their stroking length and to increase their SR to maintain their swimming velocity [2,13,22,23]. These modifications are better handled by swimmers of higher technical ability, for whom changes in the SR represent lower changes in metabolic power [15].

There are very few precedents in the literature of the stroke mechanics of open water swimmers. In a 25 km trial performed in a swimming pool, velocity and stroking length values decreased, whereas SR values were maintained at steady levels [24]. In open water conditions, the SR was augmented in the last 1000 m of a maximal 5 km test that simulated the pace of a typical race [25]. However, no additional information about how fatigue could affect SR in open water disciplines (open water, triathlon, etc.) has been found in the literature that could explain the mechanisms behind pacing strategies. Therefore, the main aim of the present research was to examine the evolution of the SR values of competitive open water swimmers according to their performance level. It was hypothesized that swimmers of a higher level would display lower average SR, but a greater ability to modify the SR according to the demands of the race.

## 2. Materials and Methods

### 2.1. Participants

All swimmers in the men’s and women’s 10 km and 25 km races at the FINA 2019 World Swimming Championships in Gwangju (South Korea) were included for analysis in the present study. The women’s 10 km was held on 14 July, the men’s 10 km was on 16 July and the 25 km (for both women and men) was on 19 July. The ambient temperature during the days and times of competition ranged from 20 °C to 25 °C, whereas the water temperature ranged from 21.8 °C to 23.8 °C, being the lowest in the 25 km race. Participants in this event were limited to only two entries per country according to FINA rules (http://www.fina.org/content/fina-rules-regulations, accessed 20 March 2020), and the 10K race served as a qualifier for Tokyo Olympics (Top-10 classified swimmers in this event automatically achieved the qualification). Final race times as well as intermediate swimming SR were collected for each athlete in the official results provided by the official timekeeper. Final times as well as intermediate race positions were collected in the open access results website (http://www.omegatiming.com, accessed 15 November 2019). Intermediate swimming SR data were provided by the official timekeeper after request. In total, 137 participants (73 men and 64 women of 23.90 ± 4.19 and 23.35 ± 3.96 years, respectively) were analysed in the 6 race laps of the 10 km race circuit, and 38 participants (22 men and 16 women of 25.12 ± 4.76 and 23.64 ± 2.69 years, respectively) were analysed in the 10 laps of the 25 km race circuit. The race circuit for the 10 km and 25 km races was different in the length of laps. The 10 km race consisted of six laps of approximately 1.666 km, whereas the 25 km race consisted of 10 laps of approximately 2.5 km. No results from the 5 km race were included for analysis, as no complete SR values through the race were provided by the official timekeeper. Only the participants who finished their races were included in the study. Additional seven participants who did not finish the race (one male in the 10 km event and two males and four females in the 25 km event) were excluded from further analysis. 

### 2.2. Data Analysis

All experimental procedures were carried out in accordance with the Declaration of Helsinki, and since the data were based on publicly available resources, no informed consent was obtained from the participants. 

Competitors were divided into different performance groups according to their finishing positions in each event, using similar criteria as previous research [4,5]. The top 10 swimmers classified were considered as successful participants, as they were granted with automatic classification for the open water event in the Tokyo 2020 Olympic Games. These swimmers were included in the G1 group. All remaining participants in the 10 km race were included in groups of similar size (11th to 25th, 26th to 40th, 41st to 55th, and 56th to 72nd for men, and 11th to 23rd, 24th to 36th, 37th to 49th and 50th to 64th for women), successively named G2, G3, G4 and G5. In the 25 km event, due to a smaller number of participants, non-successful swimmers were included in the same G2 (11th to 22nd for men and 11th to 16th for female) group.

The stroking rate values were obtained through accelerometers included in the timing bracelets (Swiss Timing, LTD) of each of the participants, which registered the number of strokes. From these data, the evolution of the swimming SR was obtained by calculating the average SR (cycles/min) per lap in each of the performance groups, as well as the overall race SR and swimming pace (time per 100 m). Dispersion in the SR values throughout the race was calculated as the coefficient of variation (%) of SR values per lap. No information about swimming paces per lap is presented, as no accurate information about the distance per lap was obtained from the Organizing Committee. Unlike traditional race circuits in open water events, lap distances in the 10 km race of the FINA 2019 World Swimming Championships did not last 2.5 km, so this event was held in six instead of the traditional four laps.

### 2.3. Statistical Analysis

A two-way (race lap × performance group) repeated-measures analysis of variance (ANOVA) was performed to compare the SR values of the different performance group, and in the case that statistical effects (*p* < 0.05) were detected, Bonferroni comparisons were performed. Effect sizes as partial eta squared (*η^2^*) values were employed to present the magnitude of differences with 0.01, 0.06 and above 0.15 thresholds for the trivial, small, medium and large effects, respectively [26]. Also, an analysis of covariance (ANCOVA) was used to determine between-group differences, using the age as the covariate. In order to assess the relationships between intermediate and overall SR values with the finishing race positions, a Pearson product–moment correlation (with 0.1, 0.3, 0.5, 0.7 and 0.9 as threshold values) [27] was employed. Variability of descriptive statistics was reported with standard deviation, and uncertainty of estimates was indicated using 95% confidence intervals (CIs). All statistical analyses were performed using IBM SPSS statistics for Windows, version 20.0 (Armonk, NY, USA: IBM. Corp.).

## 3. Results

The overall stroking values of open water swimmers in the 10 km and 25 km races of the FINA 2019 World Swimming Championships (Table 1) did not show general differences (*p* > 0.05) according to the participants’ finishing positions. However, there were trivial differences between genders, as female swimmers presented greater average SR than males (10 km: 1.88 cycles/min; 95% CI 0.56 to 3.20, *F*_1_ = 7.90; *p* = 0.01; *η*^2^
*=* 0.06 and 25 km: 2.13 cycles/min, 95% CI −0.05 to 4.30, *F*_1_ = 3.98; *p* = 0.05; *η*^2^
*=* 0.11). Between events, female swimmers of the 10 km race presented a greater SR (2.52 cycles/min, 95% CI 0.29 to 4.76, *F*_1_
*=* 8.17; *p* = 0.01; *η*^2^
*=* 0.10) than those of the 25 km race, whereas male swimmers of the 10 km and 25 km races did not show differences (*p* = 0.06). Neither the mid-race nor the overall SR values of the 10 km and 25 km races presented any large correlation with the finishing positions of the different performance groups.

For the swimming paces, G1 swimmers in the 10 km event did not show faster velocities than swimmers of a lower performance level, except between G4 and G5 swimmers in the men’s race and among G3, G4 and G5 swimmers in the women’s race (Table 1). However, in the 25 km race, successful swimmers presented faster average race velocities than non-successful swimmers, regardless of the gender (Table 1). Swimming paces in the 10 km race were considerably faster than in 25 km both for males (*F*_1_
*=* 789.20; *p* = 0.001; *η*^2^ = 0.90) and females (*F*_1_ = 489.55; *p* = 0.001; *η*^2^
*=* 0.87), regardless of the swimmers’ level.

The mid-race SR during the 10 km race of the FINA 2019 World Swimming Championships (Figure 1) varied according to the race lap (*F*_2.94_ = 59.44; *p* = 0.001; *η*^2^ = 0.33), although changes depended on the swimmer’s finishing positions (*F*_11.76_ = 16.48; *p* = 0.001; *η*^2^ = 0.36) and, in a lesser magnitude, on the swimmer’s gender (*F*_2.94_ = 2.88; *p* = 0.04; *η*^2^ = 0.02). Both male and female competitors of the first performance groups (G1 to G3) maintained their SR values on the first three laps of the race (*p* > 0.05 between laps) and from that point on, they generally increased their SR to the end of the race (maximum ∆ in the men’s race: 1.63 cycles/min, 95% CI 0.77 to 2.49 cycles/min, *p* = 0.001 from lap 3 to 4; maximum ∆ in the women’s race: 2.69 cycles/min, 95% CI 1.75 to 3.73 cycles/min, *p* = 0.001, from lap 5 to 6). Swimmers from the two slowest performance groups, on the other hand, did not increase their SR from the beginning to the end of the race. The coefficient of variation (%) of SR values decreased from G1 to G4 (3.69%, 2.54%, 2.37%, 0.91% for men and 5.01%, 3.53%, 1.85%, 1.47% for women), although the slowest performance group (G5) showed greater variation than G4 (1.68% both for men and women).

In the 25 km race (Figure 2), the SR values also varied during the race (*F*_4.77_ = 11.19; *p* = 0.001; *η*^2^ = 0.26), and their evolution depended on the swimmers’ finishing positions (*F*_4.77_ = 8.81; *p* = 0.001; *η*^2^ = 0.22)—but, most importantly, on gender (*F*_4.77_ = 25.24; *p* = 0.001; *η*^2^ = 0.44). Successful male swimmers (G1) presented several SR changes throughout the race: values increased from the 1st lap to the 2nd lap (∆: 3.36 cycles/min, 95% CI 2.17 to 4.55 cycles/min, *p* = 0.001), decreased from the 4th lap to the 5th lap and from the 5th lap to the 6th lap (maximum *∆*: −2.27 cycles/min, 95% CI −1.17 to −3.77 cycles/min, *p* = 0.001) and then increased again from the 7th lap to the 8th lap and from the 9th lap to the 10th lap (maximum *∆*: 3.73 cycles/min, 95% CI 1.64 to 5.83 cycles/min, *p* = 0.001). However, non-successful male swimmers (G2) increased their SR in each lap of the first half of the race (maximum *∆*: 2.69 cycles/min, 95% CI 1.55 to 3.83 cycles/min, *p* = 0.001, from the first to the second lap), and then, no further increases were observed to the end of the race (*p* > 0.05). Female competitors, on the other hand, decreased their SR from the first to the second lap regardless of their finishing positions (maximum *∆*: −4.10 cycles/min, 95% CI −2.75 to −10.95 cycles/min, *p* = 0.001, for the non-successful group) and then maintained their SR values for the remaining race distance (*p* > 0.05). The coefficient of variation (%) of SR values was greater for successful compared to non-successful male swimmers (7.09% vs. 4.82%), whereas for women, successful competitors showed lower variation than non-successful competitors (1.56% vs. 3.15%).

## 4. Discussion

The main aim of the present research was to examine the SR values of open water swimmers during a world level event, in order to provide some insights on how stroke mechanics contribute to the competitors’ race strategies. The results indicated that in both the 10 km and the 25 km events, SR values showed variations throughout the race that depended on the swimmer’s performance level and on gender. However, faster swimmers in the open water events did not present different overall SR values compared to swimmers of a lower expertise level.

Successful participants in the 10 km and 25 km events of the 2019 FINA World Swimming Championships displayed average swimming paces ranging from 64 to 70 and from 68 to 74 s per 100 m, in the men and women’ races, respectively. These paces were considerably faster than those reported in the 2015 and 2017 World Championships [4,5], and no statistical differences in velocity were observed when comparing successful to non-successful swimmers (at least in the 10 km race), which highlights the event’s competitiveness as well as the importance of race tactics in a mass-start discipline like open water swimming [2,28,29]. Unfortunately, in the present research, no accurate information about the lap distances (especially the first lap where swimmers begin from a starting pontoon and the last lap where swimmers arrive at the finish line) was obtained from the Organizing Committee, and consequently, no pacing variations throughout were examined. Researchers should be aware that this represents a major limitation in some previous research on open water pacing profiles [3,6], as, if lap distances of the first and last lap circuit are not corrected, the average velocities could be over- or under-estimated.

In order to attain the swimming paces referred to above, successful swimmers in Gwangju displayed average SR ranging from 34 to 36 and from 37 to 41 cycles/min in the men’s and women’s races, respectively. These values are greater than the only precedent results in the literature that examined the stroking parameters in a swimming pool 25 km time trial [24] or in open water conditions [25] during long-distance efforts and represent swimming intensities above the anaerobic threshold of international-level swimmers, specifically around the fifth and sixth step of an incremental 7 × 200 m test [30]. The swimmers’ performance level did not affect the overall SR values, although faster swimmers would be expected to display greater SR than slower ones (due to their faster velocities). A possible explanation is that perhaps swimmers of a greater expertise could achieve a certain swimming velocity with a lower SR than slower swimmers (due to a greater technical ability) [15]. However, additionally, velocities can be produced under different combinations of SR and length in participants due to differences in anthropometric characteristics, stroking technique, muscle flexibility and coordination [31]. Indeed, in the present data of the 2019 World Swimming Championships, no large correlations were observed between the partial or overall SR values and the finishing positions of the participants (both in the 10 km and in the 25 km races). Between genders, the differences in average SR were trivial—in line with previous data on different distance events [17,18], although female swimmers tended to show greater SR. It has been also acknowledged that, due to a shorter surface area of propulsive segments [18], generally observed in females, they present a greater dependence on SR to swim fast [19]. Also, the age of the participants was examined in relation to the SR values displayed, but no statistical effect was observed (in line with previous longitudinal observations by Vorontsov and Binevsk [20]).

The lap-to-lap SR values in the 10 km race of the World Championships indicated an even profile in the first half of the race (laps 1 to 3), regardless of performance level or gender. This is in line with the conservative pacing strategy previously described in 10 km elite races [3,4,5,6], which highlights the importance of swimming efficiency (lower energy cost from a certain swimming pace) to spare energy substrates and delay fatigue [13]. However, from the third lap of the circuit, there was a 1.5–2.5 cycles/min increase in the swimming SR of G1 to G3 swimmers, which was not observed in swimmers of a lower level (Figure 1). Probably, swimmers of a greater expertise employed a lower energy cost to modify their SR [15]. The maximum SR increase was observed in the last race lap for the successful female swimmers, who achieved values around 45 cycles/min and 12% greater SR that at the race beginning. As female swimmers generally present a smaller propulsive surface area [14], they rely more than males on SR to swim fast [19], which could explain their stroking dynamics at the end of the 10 km race. Successful male swimmers, on the other hand, also performed the greatest change in SR between performance groups, but G2 and G3 swimmers achieved greater SR values in the last race lap. This could highlight that an increased stroke frequency does not necessarily represents a greater propelling efficiency [32], especially for swimmers with lower levels of expertise. It should be also noted that G1 male swimmers presented their maximum increase in the SR from the third to the fourth lap rather than in the last lap, where SR could be expected to assist in the end spurt towards the finish line. Therefore, other mechanisms, like kicking action, may have been employed by them to increase their swimming pace [33]. 

For the 25 km event, the SR dynamics presented some differences in male and female swimmers; differences were observed also in the 10 km event. In the men’s race, both successful and non-successful swimmers tended to increase their SR values in the first laps, from 30 cycles/min to approximately 35 cycles/min, but, at the half-way point of the race, they decreased their SR, which was probably linked to them slowing down their swimming pace (Figure 2). A greater pacing variation was previously reported [3,4] in the 25 km event because competitors follow the behaviour of the surrounding group (during the five-hour race), rather than taking rational decisions based on their physiological status [10]. In this vein, changes in the SR profile could be expected according to mid-race pacing variations. For example, the coefficient of variation in the SR during the men’s 25 km (7.09%) was greater than in the 10 km race. However, what distinguished successful and non-successful male swimmers in the 25 km race was the ability of G1 swimmers to increase their SR in the second half of the race. While most swimmers maintained SR values between 35 and 37 cycles/min, successful swimmers progressed to SR of about 39 cycles/min (30% greater than the race lap 1), which could be related to a greater anaerobic reserve [34] or a greater aerobic capacity [35]. In the women’s 25 km race, on the other hand, swimmers presented an even SR profile for most of the race (with a lower coefficient of variation than men), and no SR differences were observed between successful and non-successful swimmers. This probably highlights the ability of female swimmers to sustain a certain swimming intensity for the entire race distance, which allows them to avoid a conservative strategy with a pacing end spurt in the last lap. Indeed, female athletes have been reported to relatively outperform their male counterparts in ultra-endurance disciplines [4,36], which—in swimming—could be supported by their body composition [37], with more buoyancy and less drag, allowing them to perform better in longer distance races [2]. Finally, it should be also acknowledged that the specific race tactics (grouping, gap times, etc.) can influence pacing (and stroking rate) variations in such a way that the trends observed may not be entirely explained by the physiological capacities of the swimmers.

Coaches should be aware that SR values of elite open water swimming races fall within 34 to 36 and from 37 to 41 cycles/min, respectively, in the men’s and women’s races. With this SR, competitors achieve swimming paces from 64 to 70 and from 68 to 74 s per 100 m of the race. Successful competitors do not generally display higher SR than non-successful competitors but they present a superior ability to modify SR throughout the race, with values that can increase up to 10% or 30% (for the 10 km or 25 km races) from the race beginning. Females seem to rely more on the SR to swim fast in open water, and this can be observed both at the end spurt (10 km) and over the entire race (25 km). Of course, further monitoring of SR is still needed in more elite races to completely understand how open water swimmers perform, but this research fills some gaps in knowledge that coaches should consider when designing training sets for event preparation. Further interpretations of stroke rate dynamics in relation to pacing laps could be done in future research, where accurate information of lap distances is provided. Caution when interpreting the results of the present research, should be taken because of the lack of information regarding 1) the anthropometric characteristics of the swimmers (that could affect stroking rate data) as well as 2) the validation of the accelerometers that registered the stroking rate values.

## 5. Conclusions

Successful open swimmers in the 10 km event of the 2019 World Swimming Championships displayed a conservative SR profile regardless of gender, with even SR (35–37 cycles/min) in the first half of the race and increases of up to 39 cycles/min in the second half. The greatest increases in SR were observed at the end of the race for successful female competitors, whereas males probably employed other propulsive mechanisms apart from the stroking rate for the race end spurt. For the 25 km race, different SR profiles were observed by gender, as successful female swimmers presented an even SR profile for most of the race (with greater overall values than males), whereas successful males presented a more variable profile throughout the race. The SR values, which were provided for the first time during 2019 World Swimming Championships, did not differentiate performance groups of swimmers nor were they related to the swimmers’ finishing positions, although they did reveal typical race profiles in line with pacing strategies of open water races.

## Figures and Tables

**Figure 1 ijerph-18-06850-f001:**
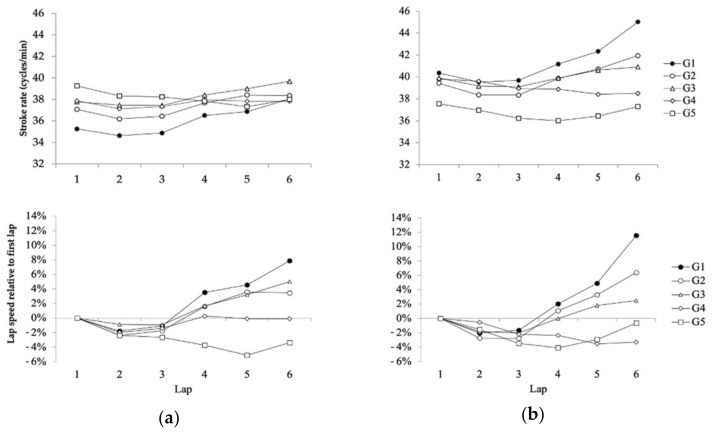
Absolute and relative stroking rate evolution of different performance group (G1 to G5) participants in the 10 km event of the FINA 2019 World Swimming Championships: (**a**) Males; (**b**) Females.

**Figure 2 ijerph-18-06850-f002:**
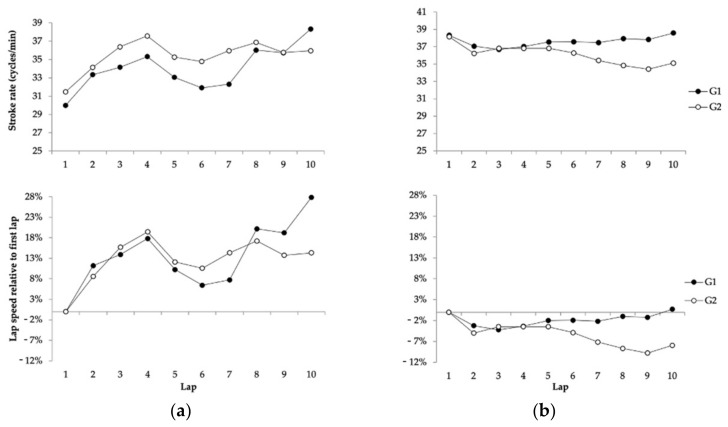
Absolute and relative stroking rate evolution of different performance group (G1 to G5) participants in the 25 km event of the FINA 2019 World Swimming Championships: (**a**) Males; (**b**) Females.

**Table 1 ijerph-18-06850-t001:** Average swimming paces and stroking rates for the different performance groups (G1: 1st–10th; G2, G3, G4 or G5: the remaining event participants divided by the number of groups) in the 10 km and 25 km open water races of the 2019 FINA World Swimming Championships.

	10 km		25 km	
	Pace (s/100 m)	SR (cycles/min)	Pace (s/100 m)	SR (cycles/min)
Male				
G1	64.80 ± 0.03	36.02 ± 4.44	70.01 ± 0.18	34.02 ± 3.65
G2	65.29 ± 0.40	37.12 ± 2.97	73.54 ± 3.54 ^1^	35.42 ± 3.22
G3	66.15 ± 0.05	38.28 ± 3.39		
G4	67.23 ± 0.56	37.65 ± 4.33		
G5	72.82 ± 2.95 ^1^	38.28 ± 3.69		
Female				
G1	68.89 ± 0.01	41.35 ± 4.02	74.51 ± 0.45	37.30 ± 3.89
G2	69.08 ± 0.15	39.78 ± 4.65	78.99 ± 4.78 ^1^	36.09 ± 3.89
G3	69.87 ± 0.85	39.93 ± 3.49		
G4	73.25 ± 1.14 ^1^	39.03 ± 3.88		
G5	80.40 ± 3.80 ^1^	36.75 ± 3.71 ^2^		

^1^ Swimming pace statistically slower than that of the previous performance group. ^2^ Stroke rate statistically slower than that of the remaining performance groups.

## Data Availability

The data that support the findings of this study are available from the corresponding and first authors (santiago.veiga@upm.es and rodriguezadalia@gmail.com) upon reasonable request.

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
