# Peer review of "Stroking Rates of Open Water Swimmers during the 2019 FINA World Swimming Championships"

_ijerph, 2021, doi:10.3390/ijerph18136850_

Round 1

Reviewer 1 Report

Line 92 – 94 -  „All swimmers in the men’s and women’s 10km and 25km races at the FINA 2019 World Swimming Championships in Gwangju (South Korea) were included for analysis in the present study.” – The date of the start was not given (10km – women – 14 July 2019, men – 16 July 2019. 25km – women, men – 19 July 2019.).

Line 95 – 96 - „Final race times as well as intermediate swimming SR were collected for each athlete from the official results provided by the official timekeeper.” - it was stated who analyzed the SR, but there was no information where the data was collected from (website, etc.)

 Line 101 – 102 – „No results from the 5km race were included for analysis, as no consistent SR values were obtained in the official race results.” – There is no description of any inconsistencies.

Line 103 – „Also, 7 participants who did not finish the race (one male in the 10km event and two males and four females in the 25km event) were excluded  from further analysis.” – it was written that both male and female competitors starting at a distance of 25 km were excluded from the analysis, while in the further part of the work (Line 116-117) competitors were included in the analysis

Line 116 – 117 – „In the 25km event, due to a smaller size of participants, non-successful swimmers were included in the same G2 (11th to 22nd for men and 11th to 16th for female) group.” – competitors who did not finish the race should be removed from the analysis.

Line 120 – The calculation procedure used to determine the SR was not stated.

The work does not specify the environmental conditions (water temperature, tides, currents, waves) that prevailed during 10 km and 25 km swimming distances. The races took place on various dates at the distance of 10 km - women - 14 July 2019, men – 16 July, 2019. and at the distance of 25 km - women, men – 19 July, 2019. Variable environmental conditions could have had an impact on SR and average swimming.

Baldassarre et al. [2017] states „Open-water races may be characterized by extreme environmental conditions (water temperature, tides, currents, and waves) that have an overall impact on performance, influencing tactics and pacing…”[1].

  1. Baldassarre, R., Bonifazi, M., Zamparo, P., and Piacentini, M. F. (2017). Characteristics and challenges of open-water swimming performance: a review. Int. J. Sports Physiol. Perform. 12, 1275–1284.

- There was no schedule of the route covered by the competitors on the 10 km and 25 km distances.

Were the laps the same or different?

The analysis of the text shows that 6 laps were made over a distance of 10 km (1 lap was 1.666 km). 10 laps were made over a distance of 25 km (1 lap was 2.5 km). It was not explained why, despite the differences in distance of 1 lap, further analysis was undertaken.

The above-mentioned inaccuracies will result in the rejection of the work.

Reviewer 2 Report

The present study aims to examine the stroking rate (SR) values of successful 15 and non-successful swimmers on the 10km and 25km races of the FINA  2019 World Swimming 16 Championships. 

It is a very interesting research to improve the knowledge on this sport modality.  However, a limitations section could be necessary.  What were the anthropometric characteristics of the swimmers? The wingspan or arm length could affect the results. I think this information should be added.

I would congratulate the authors for their work.

Reviewer 3 Report

Interesting work on the performance analysis of open water swimmers. This work follows on from works already published by the group.

An indication of how SR is measured, and the fidelity and resolution of the system used by FINA would be welcome.

Please correct the sentence in line 272 on page 7

Reviewer 4 Report

This article presents the stroke rate data of open water swimmers at the last World Championships. This aspect is really interesting and to my knowledge has never been presented in the literature before.

The abstract is very well written and structured

L41-45 : it would be interesting to highlight the swimming pace, with a reference on intensity, speed, physiological point (as lactate threshold or something like that).

L55-58 : maybe a reference on the recent paper (Drigny 2021, risk factors, etc…) here could explain more in details the impact of swimming a 25km from a physiological point of view

L59-L61 : it will be helpful to describe the material required to collect those data

L72 : I think there is a problem here. Please clarify (strokes, cycles, etc…).

L71-78 : I think you can add the reference from Seifert 2010 (Swim specialty, etc…) and talk about the differences between swimming distances.

L79-89 : I like this paragraph but maybe it would be important to describe that SR is not only depending on pacing strategy but it could also be a consequence of fatigue and biomechanics…

L92 : maybe you can detail that the event served as a qualifier for Tokyo Olympics…

L106-108 : I dont really know where this data is available… I think it is important to mention the device which allows thi data collection (brand, etc…). Or at least, mention that it was placed on the wrist of each swimmer, etc… Moreover, it is important to mention that that device is probably not validated on the literature. And later on the manuscript, that the results could be taken with precaution.

L109-118 : the groups classification deserves to be more detailed and explained. It is okay for the G1 for 10km but then we do not understand well the choices you have taken.

L119 : here you are talking about cycles/min. It is appreciated but please be consistent with the vocabular used in the introduction part.

L144-150 : I understand the results but I think it would be better to present average SR for men and women than presenting cycles differences. For example you can tell 38 cycles/min vs 39 cycles/min…

I don’t think you can write swimming cadence. Please be consistent using stroke rate.

L158-164 : There is a problem here because I observe that it is not possible to only describe overall pacing values for the different groups. As you mentionned on your introduction, pacing is very different during an open water race and the second part is swam faster than the first. So maybe, analysis (or at least a description) of the pace from final laps could be very useful. I think the paper has here the opportunity to explain much better the differences on stroke rate variations, using the pacing laps. (it doesn’t matter if laps did not have the same distance because the important thing is to compare the paces between swimmers and not between laps). This is your figure 1 but it appears that it is not very described and then discussed on your manuscript.

L211-213 : based on your data, I don’t have the same conclusion. About overall values, we agree. But maybe you can mention that successful swimmers have the capacity to increase stroke rate, more than the others ?

L221-227 : I don’t know if this paragraph is useful. Maybe on the limits section. Meybe it would be better to question the overall pace or something else. Or directly the other paragraph…

L228-251 : two points :

  • Maybe you can less affirmative about the fact that is no SR differences between level. Because I don’t see differences in absolute values for males but I see a « SR reserve » for G1 and maybe a capacity to have more economy in the 2/3 first laps.
  • And maybe explain more in details why we see more differences in females (it is well explained in the next part)

L264-268 : not sure about your interpretation. You talk about absolute values and then you talk about variations… Please clarify.

L289 : not sure if it is a better anaerobic reserve or a better aerobic capacity… Maybe you can mention it if you agree and find references

L292 : I don’t know how you can mention it but I remember well that female race and I can assume that your results are very influenced by the race strategy from 4 swimmers. I mean that those swimmers choose to have a fast start and maybe that fact had a big impact on the stroke mechanics on the rest of the race.. Maybe it could be useful to recall that every race is different and the results observed here (especially the comparison of gender) aren’t depending only a gender differences (physiologically).

L302-305 : yes !!!

L315-317 : Maybe you can be less affirmative about the male conclusion… You decide.

Round 2

Reviewer 1 Report

Thank you for correcting my comments in the article. Congratulations
